# Semaphorins and Their Roles in Breast Cancer: Implications for Therapy Resistance

**DOI:** 10.3390/ijms241713093

**Published:** 2023-08-23

**Authors:** Radhika Aiyappa-Maudsley, Louis F. V. McLoughlin, Thomas A. Hughes

**Affiliations:** 1School of Medicine, University of Leeds, Leeds LS9 7TF, UK; r.aiyappa-maudsley@leeds.ac.uk (R.A.-M.); l.f.v.mcloughlin@leeds.ac.uk (L.F.V.M.); 2School of Science, Technology and Health, York St John University, York YO31 7EX, UK

**Keywords:** breast cancer, drug resistance, immune evasion, neuropilin, plexin, semaphorins

## Abstract

Breast cancer is the most common cancer worldwide and a leading cause of cancer-related deaths in women. The clinical management of breast cancer is further complicated by the heterogeneous nature of the disease, which results in varying prognoses and treatment responses in patients. The semaphorins are a family of proteins with varied roles in development and homoeostasis. They are also expressed in a wide range of human cancers and are implicated as regulators of tumour growth, angiogenesis, metastasis and immune evasion. More recently, semaphorins have been implicated in drug resistance across a range of malignancies. In breast cancer, semaphorins are associated with resistance to endocrine therapy as well as breast cancer chemotherapeutic agents such as taxanes and anthracyclines. This review will focus on the semaphorins involved in breast cancer progression and their association with drug resistance.

## 1. Introduction

Breast cancer (BC) is a heterogenous disease that accounts for 15% of all new cancer cases in the UK. It is a leading cause of cancer-related deaths in women, with ~11,400 deaths reported every year [1]. The disease is classified into different molecular subtypes (luminal, HER-2 enriched or basal-like) or clinical subtypes (oestrogen receptor (ER), progesterone receptor (PgR) positive, human epidermal growth factor receptor-2 (HER-2) positive or triple negative (TN, ER/PgR/HER2 negative)), which were uncovered by comprehensive gene expression profiling [2]. These classifications are clinically important as they warrant distinct treatment plans; consequently, treatment follows a multimodal approach where patients are given a combination of surgery with radiotherapy, hormonal (ER positive/luminal), molecularly targeted (HER-2 positive) or cytotoxic (TNBC/basal) therapies [3]. Despite this stratified approach to treatment, patients experience therapy resistance and progression to metastatic disease. Whilst patients diagnosed with early-stage breast cancer are associated with favourable prognoses and have cure rates of 70–80%, clinically advanced BC with distant metastasis is aggressive and terminal with the currently available treatments. BC treatment is further complicated by the very wide range of molecular alterations in cancer genes and associated signalling pathways, which are regulated by the expression of numerous growth factors and receptors, secreted cytokines and proteins that drive BC progression.

## 2. Semaphorins

Semaphorins (SEMAs) are a large family of more than 21 proteins that provide chemotactic signals for cell migration and were initially identified through their roles in nervous system development. They have also now been shown to have roles in the development and maintenance of the cardiovascular, reproductive, endocrine, immune, respiratory and musculoskeletal systems. Semaphorins are characterised by the presence of a single, cysteine-rich, ~500-amino-acid extracellular domain, termed the SEMA domain, and are divided into eight classes (classes 1–7, and class V) based on their structural and sequence similarity, but only classes 3–7 are expressed in humans (Figure 1). Classes 3 and 4 each consist of seven proteins, called SEMA3A through to SEMA3G, and SEMA4A to SEMA4G, respectively. Class 5 consists of only two members (5A and 5B), class 6 consists of four (6A–D) and class 7 has a sole representative (7A). Semaphorins can be secreted (class 3), transmembrane (class 4) or glycosylphosphatidylinositol (GPI) linked (class 7) [4]. The main receptors for semaphorins are neuropilins (two different genes: NRP-1 or NRP-2) and/or plexins (four different classes: PlexinA to PlexinD), which bind to the highly conserved SEMA domain [5]. Confusingly, SEMA domains can also be found on plexins as well as some other tyrosine kinase receptors. Some semaphorins can function as receptors themselves, signalling directly from their intracellular domains when a ligand binds to the SEMA domain (reverse signalling). SEMAs can therefore not only act as ligands for plexin receptors, but also as receptors for plexin ligands [6]. In this way, semaphorins can trigger the activation of multiple signalling pathways through their ability to signal in an autocrine, paracrine and juxtacrine manner, influencing tumour angiogenesis, growth and metastasis. Some of the semaphorins are tumour suppressors with antiangiogenic and antimetastatic roles, whilst others promote tumour progression [7]. This review will focus on the various semaphorins involved in BC pathogenesis and their associations with the therapy response.

## 3. SEMA3A

Semaphorin 3A (SEMA3A) is involved in the development of the nervous system and was one of the first semaphorins identified in vertebrates [8]. SEMA3A is a candidate tumour suppressor gene that regulates the antitumour response of the immune system. Tumour-cell-derived SEMA3A was identified to inhibit the proliferation of protumoural M2 macrophages and promote the proliferation of antitumoural M1 macrophages, thereby recruiting and activating NK and cytotoxic CD8+ T cells to tumours [9]. Reduced levels of M2 macrophages are associated with reduced immune suppressive factors, including transforming growth factor-beta (TGF-β) and interleukin-10 (IL-10), and therefore tumour cells are able to evade immune responses. Increased levels of M1 macrophages can induce apoptosis and suppress the growth of tumour cells through the secretion of pro-inflammatory factors including nitric oxide (NO), IL-12 and tumour necrosis factor-alpha (TNF-α) [10,11]. The antitumour effect of SEMA3A was defined in an experimental mouse breast cancer model, using BALB/c mice transplanted with 4T1 cells overexpressing SEMA3A, compared to control 4T1 cells. SEMA3A overexpression was associated with an increase in the recruitment of CD49b+ NK cells and CD3^+^ T cells to the tumour and a higher ratio of CD8+ to CD4+ T cells, and with a significant reduction in tumour volume (61%) and weight (60%) compared to control tumours [9]. These observations were further verified in human breast cancer tissue, where positive correlations between SEMA3A expression and CD8+ T cells or NK cells were seen [12].

Apart from regulating the immune response, SEMA3A executes its tumour suppressor function within the cancer cells themselves by regulating downstream signalling through PTEN, FOXO3A and MelCAM, which have been reported to function as tumour suppressor genes themselves. To determine the effects of SEMA3A on BC growth, MDAMB-231 cells stably transfected to overexpress SEMA3A were injected orthotopically into female NOD-SCID mice. These experiments demonstrated that SEMA3A-overexpressing cancer cells demonstrated reduced breast tumour growth due to the higher activity of PTEN and MelCAM and reduced expression of vascular endothelial growth factor (VEGF) and phosphorylated FOXO3A. Loss of SEMA3A was also associated with higher-grade (grade III) ductal carcinomas and poor survival in a panel of 2878 breast cancer patients [13], associating the loss of SEMA3A with aggressive breast cancer.

## 4. SEMA3B and SEMA3F

The genes for semaphorins 3B and 3F (SEMA3B and SEMA3F) are located at chromosome 3p21.3, which is a site of frequent allelic loss and acquired promoter methylation in breast cancer, resulting in the loss of gene expression, implicating them as tumour suppressors [14]. It has been reported that re-expressing 3p21.3 inhibits tumour growth and induces apoptosis in BC due to the re-expression of SEMA3B and 3F [15], emphasising their roles in inhibiting tumour progression. SEMA3B is a downstream target of *GATA3*, an essential gene in maintaining mammary gland homeostasis and luminal cell differentiation, and together they act as potent suppressors of breast tumour growth [16]. Loss of GATA3 occurs frequently in BC and is associated with the aggressive growth and spread of the disease [17], and studies have demonstrated that overexpressing SEMA3B in GATA3-deficient BC cells replaces some aspects of GATA3 function, resulting in increased cancer cell survival. Furthermore, SEMA3B and its receptor, NRP-1, downregulate phosphatidylinositol-3-kinase (PI3K)/AKT signalling in breast cancer cells to restrict proliferation and inhibit tumour growth [7]. The PI3K/AKT pathway is involved in cell proliferation, survival and motility and is the most frequently mutated pathway in various cancers, including breast [18]. SEMA3B also promotes cancer cell apoptosis by increasing cytochrome c release and caspase-3 cleavage and inactivates proapoptotic proteins such as glycogen synthase kinase-3β, forkhead in rhabdomyosarcoma (FKHR) and mouse double minute 2 homolog (MDM-2) [19].

By contrast, SEMA3F and its receptor, NRP-2, have roles in cell migration; for example, SEMA3F gradients had repulsive effects on the migration of thymocytes [20] and the highly motile C100 breast cancer cells [21]—an effect that was blocked by an anti-NRP-2 antibody. A mechanistic explanation has been provided in the ovarian cancer cell line, OVCA, where the overexpression of either SEMA3B or SEMA3F caused the reduced expression of matrix metalloproteinases (MMP-2/9) and reduced adhesion to extracellular matrix proteins such as fibronectin, collagen type 1 and laminin [22], all of which act to reduce migratory activity [23]. Similar findings have been described in the context of prostate cancer, which is of particular relevance since—as with BC—it can be hormone regulated. In prostate cancer, SEMA3F inhibits P27 to increase the expression of E-cadherin and integrin αvβ3 to promote an anti-migratory phenotype and inhibit metastasis. SEMA3F overexpression was also associated with improved sensitivity to 5-fluorouracil (5-FU) and apoptosis [24]. A further potential mechanism by which the semaphorins impact cancer biology is through the overlap in their receptor use with the VEGF family. VEGFs have their own specific receptors (VEGFRs) but can also bind to NRP-1 and NRP-2 and have a well-established role in tumour growth by promoting neovascularisation, which is essential for the survival of the growing tumour mass. VEGFs are overexpressed in breast cancers and are associated with aggressive tumour characteristics [25]. During cancer progression, increased VEGF_165_ expression can competitively block the binding of SEMA3B (and SEMA3A) to neuropilins, thereby enhancing breast cancer cell survival by maintaining a constitutive elevation in PI3K activity and angiogenesis, while SEMA3A and SEMA3B can competitively block the binding of VEGF_165_ to the neuropilins, inhibiting angiogenesis and metastasis [26]. Similarly, in ovarian cancer, a high VEGF to SEMA3A ratio is associated with poor overall survival [27].

## 5. SEMA3C

Semaphorin 3C (SEMA3C) expression has been demonstrated to be unfavourable in terms of prognosis across various cancers, including breast [28]. Increased levels of SEMA3C transcripts were observed in poorly differentiated or advanced breast cancers and in patients who developed local recurrences. SEMA3C expression was also significantly lower in ER-positive tumours compared to ER-negative tumours, further supporting the positive association between SEMA3C and aggressive breast cancer biology [29]. The mechanism may, again, relate in part to cell adhesion and migration, since the knockdown of SEMA3C, using anti-SEMA3C hammerhead ribozyme transgenes in two different BC cell lines, decreased cell attachment to the Matrigel basement membrane, reduced proliferation and caused a 30% reduction in invasion in transwell assays [23]. Similarly, in prostate cancer, increased expression of SEMA3C in cancer cells was shown to upregulate epithelial to mesenchymal (EMT) markers, which was associated with increased migration and invasiveness and with increased tumour formation in mouse models [30].

## 6. SEMA3E

Semaphorin 3E (SEMA3E) has also been reported to promote tumour initiation or progression in BC [31]. Two distinct mechanisms have been identified. Firstly, SEMA3E, through its receptor, PlexinD1, promotes the proliferation, migration and invasion of cancer cells through its ability to activate the oncogenic signalling of epidermal growth factor receptor (EGFR) and HER-2 [32]. Secondly, SEMA3E induces the nuclear translocation of the *Snail2* transcription factor to induce EMT in tumour cells [33,34]. In a panel of 68 primary breast cancer biopsies, it appeared to be the overexpression of SEMA3E but not the PlexinD1 receptor that drove BC progression, since SEMA3E mRNA levels were significantly higher in patients with distant breast cancer metastasis at diagnosis compared to those patients whose tumours were localised to the breast, whereas PlexinD1 levels were similar in both. Focusing on the PlexinD1 receptor in breast cancer, PlexinD1 was shown to drive a significant increase in caspase-dependent cell death that was inhibited by recombinant SEMA3E, suggesting that SEMA3E’s potential oncogenic role may be in part through halting apoptosis induced by unliganded PlexinD1. In the absence of ligand binding, PlexinD1 interacts with cytoplasmic NR4A1, which triggers mitochondrial apoptosis through the release of cytochrome *c* and caspase-9 activation. This pro-survival role for SEMA3E was confirmed by the finding that the knockdown of SEMA3E induced significant cell death in the mouse mammary cell line 4T1. The authors further went on to develop a compound, SD1, consisting of the SEMA domain of human PlexinD1 protein to act as a ligand trap to sequester SEMA3E and demonstrated that sequestering SEMA3E triggered the death of PlexinD1-expressing cancer cells in vitro, thereby demonstrating the therapeutic potential of directly targeting the SEMA/Plexin axis [31].

## 7. SEMA4A

Semaphorin class 4A (SEMA4A) is a transmembrane semaphorin that was reported to be overexpressed in the serum and tissue of breast cancer patients compared to their respective normal controls [35,36]. Hypoxia (<1–2% O_2_) occurs in approximately 25–40% of breast cancers and is associated with aggressive cancer progression and therapy resistance. Liu et al., 2019 demonstrated that the expression of SEMA4A could be induced in response to hypoxia (1% and 0.2% O_2_) in MCF7 and MDA-MB-231 cells, by the binding of the hypoxic regulator, hypoxia inducible factor-1α (HIF-1α), to the promoter region of the *SEMA4A* gene. Silencing of HIF-1α was able to suppress the basal expression of SEMA4A in both cell lines. Furthermore, siRNA knockdown of SEMA4A inhibited the expression of hypoxia-regulated genes such as *VEGF* and the phosphorylation of MAPK, AKT and STAT3, whereas adding recombination SEMA4A had the opposite effect [36]. The expression of SEMA4A in hypoxic TNBC cells was associated with apoptotic resistance and increased cell proliferation [35]—all of which suggest that the expression of SEMA4A is protective to hypoxic breast cancer cells and confers a greater survival advantage against the toxic effects of hypoxia. More recently, Paranthaman and Veerappapillai developed a peptide-based vaccine that targets SEMA4A in TNBC cells [37]. Although further validation of this peptide is required both in vitro and in vivo, the development of such peptides may help in overcoming the aggressive nature of hypoxic breast cancer cells.

## 8. SEMA4B

Semaphorin 4B (SEMA4B) has been implicated in lung cancer as both a potential oncogene [38] and tumour suppressor [39]. In breast cancer, the only published observations relate to a circular RNA expressed from the gene, termed *circSEMA4B*, which encodes a novel truncated form of the protein, termed SEMA4B-211aa. *CircSEMA4B* was found to be significantly downregulated in BC tissue compared to adjacent normal tissue, and low expression of *circSEMA4B* within the BCs was positively correlated with lymph node positivity, tumour size and metastasis, implicating at least this circular transcript as a tumour suppressor in BC. Overexpression of *circSEMA4B* in MCF7 and MDA-MB-231 cells decreased proliferation compared to controls, while the siRNA knockdown of specifically the *circSEMA4B* transcript, targeting the back-splice junction sequence, increased the migration and invasive capacity of MDA-MB-231 cells. Two potential mechanisms were identified: SEMA4B-211aa can act by binding to PI3K and inhibiting its downstream oncogenic signalling, while *circSEMA4B* can act as a sponge for the microRNA miR-330-3p, thereby derepressing the expression of the microRNA’s target PDCD4, leading to the repression of oncogenic AKT signalling [40].

## 9. SEMA4C

Semaphorin 4C (SEMA4C) and its corresponding receptor, PlexinB2, are overexpressed in breast cancer tissue, with implications for both cancer progression and metastasis, as well as resistance to hormonal and chemotherapy. High expression of SEMA4C transcripts was associated with relative cancer aggression, as evidenced by shorter survival in an analysis of 1402 breast cancer patients from The Cancer Genome Atlas (TCGA) cohort. In a large multicenter retrospective cohort study consisting of 6213 patients, Wang et al., 2021 reported that serum SEMA4C was present in 84.4% of breast cancer patients, compared to only 20.75% in other solid tumours, thereby highlighting its potential use as a diagnostic biomarker in BC [41]. In vitro, the knockdown of SEMA4C or the PlexinB2 receptor in BC cell lines representing the main molecular subtypes (MCF7, MDA-MB-231 and SKBR3) was associated with the inhibition of proliferation, an effect not observed in the normal mammary epithelial cell line, HMEC-hTERT. Live cell imaging of the SEMA4C knockdown cells demonstrated that cells showed features of senescence (enlarged, flat morphology; expression of senescence-associated β-galactosidase), while cell cycle analysis confirmed a G2/M transition block, associated with the reduced expression of cell cycle regulators CCNA2 and CDK1. Overexpression of SEMA4C led to increased migration and invasion that was dependent on the downstream expression of Rho A, which has previously been associated with breast cancer metastasis. In support of this, Liu et al. demonstrated that SEMA4C expression is negatively regulated by miR-138, with the overexpression of miR-138 capable of inhibiting the invasive behaviour of breast cancer cells and reversing EMT features [42]. The role of SEMA4C in metastasis was further confirmed in vivo, where the overexpression of SEMA4C in MCF7 cells significantly increased the rate of lung metastasis in a mouse model [43]. In a further study, similarly, SEMA4C knockdown in MDA-MB-231 cells reduced tumour volumes and instances of lung and liver metastasis and also decreased macrophage infiltration, suggesting that SEMA4C may modulate immune recognition [44].

SEMA4C is also associated with resistance to commonly used breast cancer therapies. High levels of SEMA4C were associated with shorter disease-free survival specifically in ER-positive patients treated with hormonal therapy, implying an association of SEMA4C with oestrogen independence. With respect to mechanisms, the authors reported that SEMA4C-overexpressing MCF7 cells grown in the absence of oestrogen supplementation grew at faster rates compared to controls both in vitro and in vivo [43]. This study also reported reduced levels of ER and PgR expression along with reduced nuclear localisation, indicating the reduced functional activity of the hormone receptors in these cells. Furthermore, the increased expression of SEMA4C has also been shown in paclitaxel-resistant BC cell lines compared to their respective parental cell lines, with the resistant phenotype being dependent on SEMA4C expression, thereby suggesting that SEMA4C contributes to resistance to cytotoxics [45].

## 10. SEMA4D

Bone is a common site of BC metastases, especially in tumours of luminal subtypes. Previous studies have demonstrated that semaphorin 4D (SEMA4D) is produced by skeletal cells such as osteoclasts and, through its receptor PlexinB1, acts on osteoblasts to prevent their differentiation and motility, subsequently inhibiting new bone formation [46]. Yang et al., 2016 established an association between SEMA4D expression and bone metastasis in breast cancer [47]. Using in vitro and in vivo experiments such as a mouse model of skeletal metastasis, the authors demonstrated that SEMA4D produced by MDA-MB-231 cells was able to halt the differentiation of the osteoblast cell line MC3T3. SEMA4D also affected the ability of these cells to form mineralised tissue. They further reported a decrease in the incidence of bone metastasis from MDA-MB-231 cells transfected with siRNA against SEMA4D. Clinically, bisphosphonates are used in the treatment of patients with bone metastasis, which is associated with skeletal fragility. Inhibiting SEMA4D in such patients offers a potential alternative, without any of the associated side effects observed with bisphosphonate treatment [47]. Pepinemab, a humanised monoclonal antibody against SEMA4D, is being investigated clinically for the treatment of various malignancies and neurodegenerative disorders. In breast cancer, a combination of pipinemab and trastuzumab is being assessed in metastatic HER2+ breast cancer (NCT05378464).

## 11. SEMA6D

Semaphorin 6D (SEMA6D) is a transmembrane semaphorin that contains the SEMA and plexin–semaphorin–integrin (PSI) domain in its extracellular region, allowing it to act as a secreted cytokine, exerting its effects both locally, through cell–cell interactions, and more distantly through the diffusion of its cleaved ectodomain. Overexpression of SEMA6D has been reported to increase the proliferation of the breast cell line MCF10A, which is not derived from cancer, while, by contrast, it decreased the proliferation of MCF7 cancer cells; however, this differential response was not maintained for migration, which was increased in both cell lines [48]. SEMA6D is reported to be also associated with the chemotherapy response in BC. We have previously shown that miR-195 and miR-26b were upregulated in breast cancer cells, demonstrating chemotherapy resistance in patients who were treated with epirubicin and cyclophosphamide. Both microRNAs target SEMA6D and therefore their upregulation was associated with the decreased expression of SEMA6D, while the knockdown of SEMA6D directly caused relative chemoresistance [49]. Furthermore, survival analysis using the Molecular Taxonomy of Breast Cancer International Consortium (METABRIC) cohort (*n* = 1979) [49] and other independent patient samples [50] showed significant correlations between low SEMA6D expression and the poor overall survival of breast cancer patients. Therefore, reduced SEMA6D levels are apparently associated with therapy resistance, although the mechanisms remain unknown.

## 12. SEMA7A

Semaphorin 7A (SEMA7A) is a GPI-linked semaphorin that can be released from cell membranes into interstitial fluid and has been reported to have pro-tumourigenic effects in BC, influencing tumour progression, lymphangiogenesis and drug resistance. SEMA7A is upregulated in BC compared to normal tissue in several independent data sets, and highly SEMA7A-expressing tumours were associated with worse overall survival and with early metastases. In vitro, the knockdown of SEMA7A was found to decrease proliferation and motility in the ductal carcinoma in situ cell line MCF10DCIS. In vivo, SEMA7A knockdown in MCF10DCIS cells led to significant decreases in tumour growth, the lower expression of invasive markers such as p63 and vimentin and the increased expression of E-cadherin, which suggest that the knockdown of SEMA7A may reverse EMT. In vivo assessment of SEMA7A knockdown in MDA-MB-231 cells showed a decrease in lung metastasis, further suggesting its role in tumour progression. Interestingly, the addition of either MCF710DCIS-conditioned media or recombinant SEMA7A alone to an in vitro lymphangiogenesis assay resulted in significant increases in the formation of lymphatic structures. Clinically, breast tumours positive for lymph vessel invasion showed higher SEMA7A expression, in the METABRIC dataset, again linking SEMA7A to lymphangiogenesis and implicating SEMA7A in the stimulation of dissemination via the lymphatics [51]. SEMA7A has also been implicated in resistance to anoikis [52], a specialised form of apoptosis that is induced in epithelial cells by the loss of attachment to substrates. Overexpression of SEMA7A induced anoikis resistance in the non-transformed MCF10A mammary epithelial cell line through the activation of pro-survival AKT signalling. This resistance was also associated with an increase in CD44+ CD24− stem-like cells, and—importantly in clinical terms—resistance to the chemotherapy agent paclitaxel [52], both of which could potentially lead to increased rates of metastasis.

SEMA7A is also hormonally regulated, as seen by its increased expression upon oestrogen treatment in luminal cell lines MCF7 and T47D. However, treatment of these cells with the ER downregulator fulvestrant was not sufficient to completely inhibit SEMA7A expression. Furthermore, long-term oestrogen deprivation in these cells increased the expression of SEMA7A compared to the wild-type controls. Such studies demonstrate that oestrogen does induce SEMA7A expression, but that SEMA7A can also be expressed independently of oestrogen stimulation. It was also shown that the dependence of the ER-positive MCF7 cells on oestrogen for growth in vivo was abolished when SEMA7A was overexpressed. Furthermore, SEMA7A-overexpressing tumours showed greater tumour growth in the presence of fulvestrant when compared to controls. These tumours demonstrated a relative increase in Ki-67 expression, decreased necrosis and an increase in lung metastasis. In the metastatic samples, SEMA7A overexpression was associated with decreased ER, further confirming ER independence, which was also associated with resistance to other endocrine therapies, such as tamoxifen, and/or aromatase inhibitors. Furthermore, T47D cells that are tamoxifen-resistant also exhibit the upregulated expression of SEMA7A. Breast cancer cells overexpressing SEMA7A were also resistant to treatment with the CDK4/6 inhibitor palbociclib [53]. In the clinical setting, high expression of SEMA7A has been significantly associated with both poor overall survival and poor distant metastasis-free survival in ER-positive breast cancer [51], indicating the importance of SEMA7A in contributing to endocrine therapy resistance and the associated poor prognosis observed in ER-positive tumours.

Finally, SEMA7A has been suggested as a biomarker for BC recurrence specifically in patients with postpartum breast cancers (PPBCs; defined as BC in women within 10 years of most recent childbirth). PPBCs have a three-fold higher risk of metastatic recurrence compared to their nulliparous counterparts; therefore, their biology may differ substantially. Studies conducted by Borges et al. reported that SEMA7A demonstrated significantly higher expression, an association with lymph node positivity, lymphovascular invasion and poorer outcomes in PPBC patients (*n* = 66), compared to nulliparous patients (*n* = 47). SEMA7A expression was also highest in relapsed PPBC patients with loco-regional or metastatic disease [54].

## 13. Therapy Resistance across Breast Cancer and Other Malignancies

At least four different semaphorins have been associated with resistance to different BC therapies, as described above; however, the mechanisms by which these semaphorins confer resistance to endocrine therapy and chemotherapy have not yet been fully elucidated. Conversely, the associations between semaphorins and therapy resistance are even more extensive when considering other malignancies (Table 1). Interestingly, individual semaphorins are often associated with resistance to multiple different therapeutics in difference cancers, underlining the complex nature of their biology.

SEMA3A

Studies have demonstrated that the treatment of a mouse model of pancreatic neuroendocrine cancer (the genetic RIP-Tag2 model) with sunitinib, a small molecule tyrosine kinase inhibitor, was effective in inhibiting angiogenesis and reducing the tumour volume. However, mice treated with sunitinib demonstrated an increase in local invasion, distant metastasis and hypoxia compared to the control mice. These unhelpful effects of sunitinib were overcome with combined SEMA3A treatment, with the combination demonstrating not only a reduced tumour burden and reduced angiogenesis but also reduced cancer invasion into surrounding tissue, including lymph node and liver metastases [55]. Combining SEMA3A treatment with sunitinib also reversed hypoxia and promoted blood vessel normalisation by reducing vascular branching [55]. In breast cancer, a phase II clinical trial evaluated the efficacy of sunitinib in metastatic breast cancer patients with favourable outcomes in TNBC and HER-2-positive tumours (NCT00471276). Combining SEMA3A with sunitinib may reduce treatment-induced hypoxia in breast and other solid cancers, which may be associated with better treatment outcomes. Such studies demonstrate the therapeutic potential of SEMA3A in overcoming the hypoxia and metastatic dissemination in cancer and the potential for its use in other malignancies. However, the tumour-promoting or -suppressing roles vary in the context of different cancer types. In prostate cancer, SEMA3A is overexpressed in tumour tissue and is associated with the recruitment of monocytes and their polarisation into M2 macrophages, which are associated with resistance to androgen deprivation therapy, whereas, in breast cancer and glioblastoma (GBM), SEMA3A recruits antitumour macrophages. SEMA3A was also shown to bind to NRP-1 in TAMs, promoting phosphorylation and the activation of the oncogenic PI3K/AKT pathway. However, when NRP-1 is downregulated in TAMs, SEMA3A/PlexinA1/PlexinA4 stops TAMs from migrating and inhibits their immunosuppressive and angiogenic properties, which impedes tumour growth [56]. Similar findings have also been reported in GBM, where the antibody blockade of SEMA3A was found to downregulate the PI3K/AKT pathway, inhibiting tumour growth and the recruitment of macrophages to the tumour site [57]. Figure 2 shows some of the pathways that are controlled by semaphorins in cancer progression and therapy resistance.

2.SEMA3B, SEMA3E and SEMA3F

SEMA3B and SEMA3E have been shown to be consistently upregulated in response to cisplatin treatment, designating them as a risk factor for acquired cisplatin resistance [58]. This association is apparently causative, since transfecting cisplatin-sensitive cells to overexpress SEMA3E also conferred a drug-resistant phenotype in the TYKnuR cell line. The increase in the expression of SEMA3E was also observed when cells were exposed to other chemotherapeutic agents such as carboplatin, adriamycin, mitomycin C, etoposide and radiation (UV and X-ray), all of which implies that cancer cells may upregulate SEMA3E as a survival mechanism to prevent cell death induced by chemotherapy or radiotherapy [59]. Several of these agents, such as 5-FU, platinum-based compounds and ionising radiation, are used in breast cancer treatment. Whilst no associations between SEMA3E and chemoresistance have yet been made in BC, this may be worth testing in patient samples. Importantly, this resistance may be therapeutically inhibited; for example, Casazza et al. demonstrated that a mutated, uncleavable version of SEMA3E (Uncl-Sema3E) bound to PlexinD1 reduced angiogenesis, growth and metastasis in mouse models transplanted with 4T1 cells and in tumours formed by the Lewis lung carcinoma cells (LLc), which are refractory to anti-VEGF treatments [60]. Combination treatments with semaphorins or antibodies against semaphorins may help to improve the therapeutic response and overcome chemoresistance in patients across a wide range of cancers.
ijms-24-13093-t001_Table 1Table 1Semaphorin-induced drug resistance or sensitisation in breast and other cancers.SemaphorinsDrug Response—Breast CancerDrug Response—Other CancersClinical TrialsSEMA3A-Chemosensitisation with sunitinib in pancreatic cancer [55]


Chemoresistance to androgen deprivation therapy in prostate cancer [56]
SEMA3B-Chemoresistance to cisplatin in endometrial cancer [58]
SEMA3C-Chemoresistance to gemcitabine in pancreatic cancer [30]
SEMA3E-Chemoresistance to cisplatin, carboplatin, adriamycin, mitomycin C, etoposide in ovarian cancer


Radioresistance to UV, X-ray radiation in ovarian cancer [59]
SEMA3F-Chemosensitisation to 5-FU in prostate cancer [24]
SEMA4A-Improves efficacy of anti-PD-1 antibody in NSCLC [61]
SEMA4CEndocrine therapy resistance [43]Chemoresistance to cisplatin in cervical cancer [62]

Chemotherapy resistance to paclitaxel [45]

SEMA4DRisk of bone metastasis [47]Chemoresistance to 5-FU in colorectal cancer [63]Pipinemab NCT05378464—Breast cancerNCT05102721—Pancreatic cancerNCT03320330—Refractory solid cancers in young patientsSEMA6DChemosensitisation to epirubicin [49]Chemoresistance to cisplatin in osteosarcoma [64]
SEMA7AEndocrine therapy and Palbociclib resistance [53]Paclitaxel resistance [52]Chemoresistance to TKIs in lung cancer [65]
Abbreviations: 5-FU—5-fluorouracil; TKI—tyrosine kinase inhibitor.


3.SEMA4A, SEMA4C and SEMA4D

In BC, SEMA4A is highly expressed in tumour tissue and is protective against apoptosis and hypoxia. However, Naito et al. demonstrated, in a retrospective multicenter cohort study, that SEMA4A-positive non-small-cell lung cancers responded significantly better to anti-programmed cell death 1 (PD-1) antibody compared to SEMA4A-negative cells. The addition of recombinant SEMA4A additionally induced the proliferation of T cells and improved the therapeutic efficacy of anti-PD-1 antibodies in vivo, suggesting that SEMA4A can be used as an important biomarker to assess the response to immune checkpoint inhibitors [61]—this has not yet been assessed in BC. In cervical cancer, miR-31-3p/SEMA4C signalling drives EMT and cisplatin resistance [62], adding weight to its involvement in endocrine and chemotherapy response in breast cancer. SEMA4D expression has been associated with bone metastasis in breast and distant metastasis across various other cancers, which led to the development of the monoclonal antibody pipinemab, currently undergoing clinical evaluation in breast, pancreatic and head and neck cancers. Pipinemab also decreased tumour volumes and improved survival rates in RIP-Tag2 mice models, although this was associated with an increase in lymph node metastasis, consistent with previous studies that found that anti-SEMA4D treatments increased TAMs [66]. One of the mechanisms by which TAMs promote metastasis is by secreting stromal-cell-derived factor 1 (SDF1), which increases tumour migration. Such studies provide insights into druggable targets and combination treatment strategies to improve the therapeutic efficacy of semaphorin-driven cancers.

4.SEMA6D

We have previously shown that miR-195 and miR-26b reduce the expression of SEMA6D in breast cancer, which is associated with anthracycline chemotherapy resistance. However, by contrast, SEMA6D overexpression is associated with resistance to cisplatin in osteosarcoma (OS) cells, which is regulated by miR-506, demonstrating that these effects may be specific to cancer types or to chemotherapy agents. In OS cells, knocking down circUBAP2 decreased SEMA6D expression, inhibited cisplatin resistance and the Wnt/β-catenin signalling pathway and decreased the proliferation, migration and invasion of OS cells—an effect that was reversed by the overexpression of SEMA6D. The authors used an miR-506 inhibitor, which increased the expression of both genes, suggesting that miR-506 was a negative regulator of both *circUBAP2* and SEMA6D [64]. This crosstalk between semaphorins and their regulatory microRNAs can also be therapeutically exploited to treat drug resistance in cancers.

5.SEMA7A

SEMA7A is reported to be hormonally regulated in breast cancer; however, SEMA7A is expressed independently of oestrogen stimulation. The authors suggested that this increase could be due to the influence of other hormone receptors, such as the androgen receptor, which is also increased in conditions of oestrogen deficit [64]. SEMA7A was identified to induce resistance to tamoxifen, fulvestrant and palbociclib, through upregulating the apoptotic regulator BCL2. The BCL2 inhibitor venetoclax was effective in decreasing the clonogenicity of breast cancer cells in vitro and reduced tumour volumes in mice overexpressing SEMA7A in vivo, when given in combination with fulvestrant. SEMA7A overexpression has also been reported to confer resistance to tyrosine kinase inhibitor (TKI) treatment in EGFR mutant lung cancers, associating SEMA7A both with TKI resistance in lung [65] and endocrine therapy resistance in breast cancer [53]. The mechanism of TKI resistance is via the activation of the downstream ERK pathway, which was confirmed using mTOR inhibitors. It may be worthwhile to investigate whether SEMA7A signals through the same pathway in breast cancer to contribute to endocrine therapy resistance. Identifying the signalling pathways responsible for semaphorin-induced drug resistance can help in devising combinatorial treatments to combat mechanisms of drug resistance. In this study, the combined treatment of TKIs with mTOR inhibitors led to a greater increase in sensitivity to TKIs than TKIs alone.

## 14. Conclusions

Semaphorins are emerging as important therapeutic targets, with studies demonstrating that the inhibition or promotion of semaphorin signalling can restrict the growth, metastasis and therapy resistance of cancer cells. Further validation of anti-semaphorin antibodies or peptide-based vaccines is required both in vitro and in vivo, if the complex nature of semaphorin-induced therapy resistance in breast and other cancers is to be targeted. Preclinical studies have shown that therapeutic intervention using SEMA3A/SEMA4D antibodies to decrease TAMs is a potential strategy for cancer treatment. Combining immunotherapy with anti-semaphorin antibodies can be another therapeutic strategy to avoid immune evasion by cancer cells. Numerous studies have shown that microRNAs are crucial regulators of semaphorin signalling—an association that can be exploited for chemosensitisation strategies. However, the intricate and complex network of semaphorin signalling is not yet fully understood. Further research is required into the mechanisms by which semaphorins induce resistance to therapy and whether this may be exploited during cancer treatment across a wide range of cancer types.

## Figures and Tables

**Figure 1 ijms-24-13093-f001:**
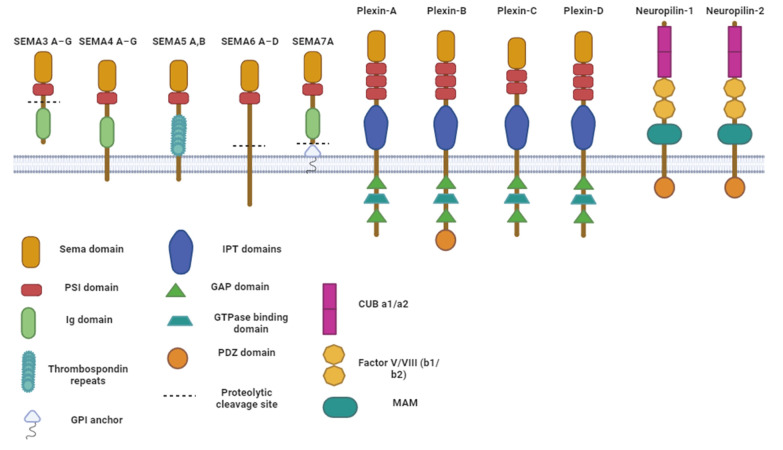
Semaphorins and their receptors. Semaphorins are divided into 8 classes. Neuropilins (1 and 2) and plexins (A to D) are the main receptors for semaphorins, which bind to the SEMA domain, activating a plethora of signalling pathways associated with proliferation, angiogenesis, the immune system, migration, metastasis and drug resistance. Abbreviations: PS1—plexin–semaphorin–integrin; GPI—glycosylphosphatidylinositol; Ig—immunoglobin; GAP—GTPase activating protein; PDZ—postsynaptic density protein of 95 kDa, Drosophila disc large tumour suppressor and zonula occludens-1 protein; OPT—IG-like plexin transcription factors; MAM—meprin/A5-protein/PTPmu.

**Figure 2 ijms-24-13093-f002:**
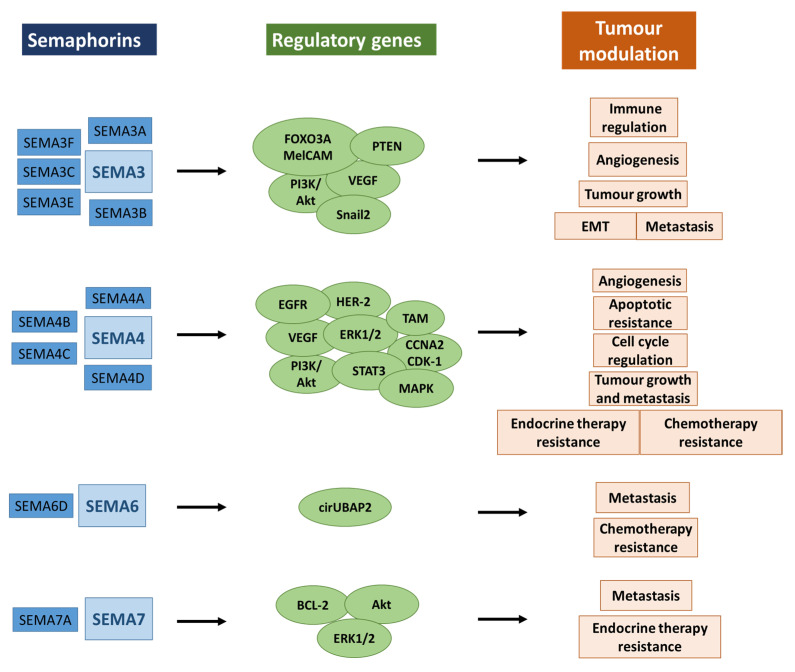
Semaphorins in cancer. Semaphorins are aberrantly expressed in cancers and regulate multiple pathways associated with tumour progression and drug resistance. Abbreviations: PTEN—phosphatase and tensin homolog; VEGF—vascular endothelial growth factor; TAM—tumour-associated macrophage; EGFR—epidermal growth factor receptor; HER-2—human epidermal growth factor receptor-2; STAT-3—signal transducers and activators of transcription 3; MAPK—mitogen-associated protein kinase; PI3-K—phosphoinositide 3-kinase; BCL-2—B-cell lymphoma-2; ERK-1/2—extracellular signal-regulated protein kinases 1 and 2.

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
