# Peer review of "Semaphorins and Their Roles in Breast Cancer: Implications for Therapy Resistance"

_ijms, 2023, doi:10.3390/ijms241713093_

Round 1
Reviewer 1 Report
This article reviews the roles of semaphorins (SEMAs) in breast cancer. Semaphorins can be divided into several classes, and the authors described each semaphorin, expressed only in humans, and summarized their functions in breast cancer. Each section, by each subclass of semaphorin, is clearly written with recent literature and is easy to follow. In the last section, the authors discussed SEMAs, which may cause therapy resistance across breast cancer and other malignancies. Table 1 summarizes well with recent studies, but unfortunately, not all SEMAs have been reported in drug response in breast cancer and other tumors. Hopefully, future research can fill in SEMA3 and SEMA4A. This manuscript is valuable since it contains helpful information for further research and clinical studies developments. I recommend publishing this manuscript in the International Journal of Molecular Sciences.
Author Response
Many thanks for your supportive comments.
Reviewer 2 Report
The article entitled "Semaphorins and their roles in breast cancer: implications for therapy resistance" by Radhika Aiyappa-Maudsley et al. is a worthwhile read. However, there are some issues that need to be addressed.
1. It would be helpful if the authors could include a figure illustrating the connection between semaphorin-induced drug resistance or sensitization in breast cancer.
2. A figure related to therapy resistance across other malignancies would also be beneficial to help readers better understand the broader implications of the findings.
3. The section on drug resistance could be expanded to provide more details on the mechanisms by which semaphorins confer resistance to endocrine therapy and chemotherapy. This would help readers better understand the clinical implications of these findings.
4. It would be beneficial if the authors could include some discussion on the potential use of semaphorins as therapeutic targets in breast cancer treatment. Additionally, it would be helpful to know if there are any ongoing clinical trials exploring this possibility.
5. The review would benefit from a more structured organization. Consider dividing the content into subsections to make it easier for readers to follow the flow of the argument.
Author Response
The article entitled "Semaphorins and their roles in breast cancer: implications for therapy resistance" by Radhika Aiyappa-Maudsley et al. is a worthwhile read. However, there are some issues that need to be addressed.
- It would be helpful if the authors could include a figure illustrating the connection between semaphorin-induced drug resistance or sensitization in breast cancer.
A figure has been added on page 9
- A figure related to therapy resistance across other malignancies would also be beneficial to help readers better understand the broader implications of the findings.
Table 1 summarises semaphorins involved across various malignancies and therefore another figure detailing this has not been added.
- The section on drug resistance could be expanded to provide more details on the mechanisms by which semaphorins confer resistance to endocrine therapy and chemotherapy. This would help readers better understand the clinical implications of these findings.
The section ‘Therapy resistance across breast cancer and other malignancies’ has been expanded and changes are highlighted in yellow
- It would be beneficial if the authors could include some discussion on the potential use of semaphorins as therapeutic targets in breast cancer treatment. Additionally, it would be helpful to know if there are any ongoing clinical trials exploring this possibility.
This has been discussed throughout the manuscript and ongoing clinical trials with SEMA4D antibody pipinemab has been discussed on page 11. Changes are highlighted in yellow
- The review would benefit from a more structured organization. Consider dividing the content into subsections to make it easier for readers to follow the flow of the argument.
The section ‘Therapy resistance across breast cancer and other malignancies’ has been subdivided into sections to make it easier for readers.
Reviewer 3 Report
The manuscript provides an overview of the role of Semaphorins in breast cancer treatment resistance. In general, it is a well-structured paper. The topic is interesting and could provide some insights for researchers. However, in view of the amount of data presented, I would suggest considering it as a short review.
Here are some other tips:
- Classification of breast cancer based on hormone receptor expression (lines 24/25) must also include those sensitive or not to progesterone.
- The work could be improved by including one or two figures (e.g., three-dimensional protein structures, schematic representations of the mechanism of action, and so on).
- The conclusions paragraph could be implemented to better emphasize the importance of the data described in the entire manuscript.
- Authors should check the formatting of all bibliographic references according to journal rules, and add some important ones such as:
Theranostics. 2021; 11(7): 3262–3277 . Semaphorins as emerging clinical biomarkers and therapeutic targets in cancer (in the Introduction);
and
Acta Physiologica, 231 (4) e13570. DOI: 10.1111/apha.13570 The chromogranin A1-373 fragment reveals how a single change in the protein sequence exerts strong cardioregulatory effects by engaging neuropilin-1. (2021).
After these modifications, the manuscript can be accepted as a short review.
Only minor English language revisions throughout the text are required.
Author Response
- Classification of breast cancer based on hormone receptor expression (lines 24/25) must also include those sensitive or not to progesterone.
The changes have been made in the manuscript and highlighted in yellow on lines 26/27/28.
- The work could be improved by including one or two figures (e.g., three-dimensional protein structures, schematic representations of the mechanism of action, and so on).
A new figure has been added on page 9
- The conclusions paragraph could be implemented to better emphasize the importance of the data described in the entire manuscript.
The conclusion paragraph has been amended and changes are highlighted in yellow.
- Authors should check the formatting of all bibliographic references according to journal rules, and add some important ones such as:
Theranostics. 2021; 11(7): 3262–3277 . Semaphorins as emerging clinical biomarkers and therapeutic targets in cancer (in the Introduction);
The above reference has been added to the introduction on line 64
and
Acta Physiologica, 231 (4) e13570. DOI: 10.1111/apha.13570 The chromogranin A1-373 fragment reveals how a single change in the protein sequence exerts strong cardioregulatory effects by engaging neuropilin-1. (2021).
The above reference has been added to the introduction on line 56
After these modifications, the manuscript can be accepted as a short review.
Round 2
Reviewer 2 Report
Accept in present form